# Influence of the maternal high-intensity-interval-training on the cardiac *Sirt6* and lipid profile of the adult male offspring in rats

**Reihaneh Mohammadkhani**[1], **Neda Khaledi**[1]*, **Hamid Rajabi**[1], **Iraj Salehi**[2☯],
**Alireza Komaki**[2☯]

**1** Department of Exercise Physiology, Faculty of Physical Education & Sports Science, Kharazmi University, Tehran, Iran, **2** Neurophysiology Research Center, Hamadan University of Medical Sciences, Hamadan, Iran

☯ These authors contributed equally to this work.
* n.khaledi@khu.ac.ir

**Data Availability Statement:** All relevant data are within the manuscript and its Supporting Information files.

## Abstract

The susceptibility to cardiovascular disease in offspring could be reduced prior to birth through maternal intervention, before and during pregnancy. We evaluated whether the initiation periods of maternal exercise in preconception and pregnancy periods induce beneficial effects in the adult male offspring. Thirty-two female rats were divided into control and exercise groups. The exercise groups involve exercise before pregnancy or the preconception periods, exercise during pregnancy, and exercise before and during pregnancy. The mothers in the exercise groups were run on the treadmill in different periods. Then the birth weight and weekly weight gain of male offspring were measured, and the blood and left ventricle tissue of samples were collected for analysis of the Sirtuin 6 (Sirt6) and insulin growth factor-2 (IGF-2) gene expression, serum levels of low-density lipoprotein (LDL), high-density lipoprotein (HDL), cholesterol (Cho), and triglycerides (TG). There was no significant difference in the birth weight of offspring groups (P = 0.246) while maternal HIIT only during pregnancy leads to reduce weekly weight gain of offspring. Our data showed that Sirt6 and IGF-2 gene expression was increased (P = 0.017) and decreased (P = 0.047) by maternal exercise prior to and during pregnancy, respectively. Also, the serum level of LDL (p = 0.002) and Cho (P = 0.007) were significantly decreased and maternal exercise leads to improves the running speed of the adult male offspring (p = 0.0176). This study suggests that maternal HIIT prior to and during pregnancy have positive intergenerational consequence in the health and physical readiness of offspring.

## Introduction

Based on cardiovascular research, the risk factors which are independent of genetics could be altered by maternal behavior during pregnancy [1]. Exercise is a widely accepted positive intervention during pregnancy, which could reduce the risk of cardiovascular disease (CVD) development before birth [2].

**Funding:** This work was supported by a grant of Hamadan University of Medical Sciences, Iran. The funders had no role in study design, data collection and analysis, decision to publish, or preparation of the manuscript.

**Competing interests:** The authors have declared that no competing interests exist.

Although previous studies have shown the positive effect of maternal exercise on vascular function and heart rate in adult offspring [3, 4], little is known about the benefits of maternal exercise during pregnancy to reduce the risk of heart disease in offspring. Brito et al. confirmed that low intensity exercise during pregnancy increases the expression of cardiac Sirtuin 6 (Sirt6) and maternal exercise has a cardioprotective effect in the hearts of progeny [5]. There is a close link between Sirtuin activity and exercise by increasing nicotinamide adenine dinucleotide ($NAD^+$) levels [6, 7]. There is evidence that Sirt6 protects the heart from developing diseases [8, 9] through negative regulation of the insulin growth factor (IGF) signaling in the myocardial cells [10]. The IGF system has an important role in cardiac growth during pregnancy and it is established that there is a link between the low birth weight and an increase IGF-2 gene expression in the heart that leads to an increased risk of cardiovascular disease in later life [11]. Indeed, the Sirt6 could be negatively regulated low-density lipoprotein (LDL) levels [12]. The proper level of serum lipid profile LDL, high-density lipoprotein (HDL), cholesterol (Cho), and triglycerides (TG) are considered as common biomarkers for cardiovascular health. Therefore, it seems that Sirt6 and blood lipid profile could be used as prediction components for cardiovascular health that could be altered by maternal exercise.

A recent study has shown that high-intensity exercise during pregnancy not only has no negative effects on maternal cardiac function, but it could also be tolerated by pregnant rats, which activated the protective mechanism in fetal offspring by increasing the total antioxidant capacity [13]. The high-intensity interval training (HIIT) is defined as a short and repeated period of intense exercise close to 85% to 95% of maximal heart rate with less intense recovery periods [13–15]. Recent studies support the notion that cardiovascular adaptations of exercise linked to improved cardiac health consequences are intensity-dependent and there is a significant difference in the cardiovascular benefits of strenuous intensity versus moderate intensity [15, 16]. Because regular physical activity in the preconception periods should be an important component of a healthy pregnancy [17], improving preconception health of mothers can result in improved pregnancy related-outcomes and offspring health.

However, we hypothesized whether the initiation time of maternal HIIT is a significant factor for health consequences in the offspring. To test the beneficial effect of maternal HIIT on the health of offspring, we evaluated the serum level of lipid profiles, the cardiac Sirt6 and IGF-2 gene expression as cardioprotection factors. Also, we measured birth weight and weekly weight gain and performed an exercise test on the male offspring.

## Material and method

### Animal

Thirty-two female *Wistar* rats which never experienced a pregnancy, aged 8 weeks, were purchased from the animal house of Hamadan University of Medical Sciences; The animals were kept in cages with 4 rats in each cage. The rats were first acclimatized to the treadmill exercise for one week before the experiment. The animals were housed in an air-conditioned room at $22 \pm 2°C$ with a 12-h light/dark cycle. Standard animal chow and water were freely available. Animal care and experimental protocols were approved by the Veterinary Ethics Committee of Hamadan University of Medical Sciences (AEC: IR.UMSHA.REC.1397.528) and The National Institutes of Health Guide for Care and Use of Laboratory Animals (NIH Publication no. 85–23, revised1985) was complied with.

### Study design

A summary of the present study is depicted in Fig 1. High-intensity-interval training was performed in two parts, the first part included six weeks HIIT prior to mating and the

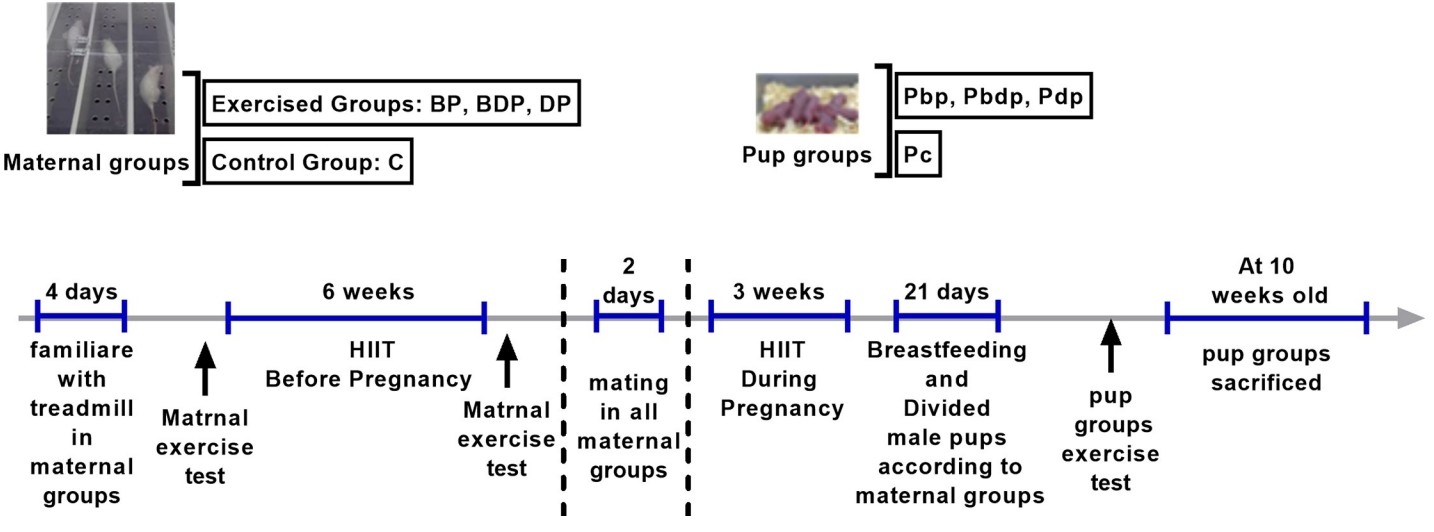

**Fig 1. Schematic experimental design of the study.**

second part included 3 weeks HIIT after mating. Animals were randomly divided into two maternal groups; maternal sedentary (control group, n = 8) and maternal exercise groups (n = 24). Maternal exercise groups were also divided into 3 subgroups (BP, BDP and DP groups) based on the initiation period of the maternal exercise; BP group included mothers who exercised only before pregnancy, BDP group included mothers who exercised both before and during pregnancy, and DP group included mothers who exercised only during pregnancy.

Prior to initiating the exercise protocol, all animals were familiarized with treadmill exercise (model: 2016 Tajhiz Gostar Omid Iranian, Iran) for four days (10 mins with 10 m/min at 0° inclination) to reduce stress. The exercise test was accessed individually for maternal groups before and after six weeks of HIIT and also for pup groups at 10 weeks old. Also, animals were known to run on the treadmill through physical handling. After the last session of six weeks exercise two females with one male were housed for two days mating and then they were separated and exercise during pregnancy was initiated. After three weeks, pregnant rats were kept individually per cage to notice the birth process, number, and birth weight of pups. some pups were given to other mothers, who were not related to present study to create uniform condition per groups and male rats were separated from their mothers at the end of the breastfeeding duration (3 weeks old) and were allocated in pup groups (**P**c, **P**bp, **P**bdp, and **P**dp); **P**c group included pups of maternal control group, **P**bp group included pups of mothers who exercised before pregnancy, **P**bdp group included pups of mothers who exercised before and during pregnancy, and **P**dp group included pups of mothers who exercised during pregnancy. The number of pups per group were based on the number of pregnant mothers. The body weight expressed in grams was recorded immediately after birth and evaluated weekly until the 8th week in the pup groups. After measuring the final body weight of pups (10 weeks old), the rats were anesthetized with Xylazine (3 mg/kg) and Ketamine (30 mg/kg), the heart was weighed, and the left ventricle was separated after whole blood extraction. Left ventricle harvest was washed with Phosphate-buffered saline (PBS); snap-frozen in liquid nitrogen and also Blood samples were centrifuged at 2500 g for 15 min; the serum and tissue aliquots were stored at -80° for more analysis.

## Mating

Two females who mate with one male were separated after two days and then the vaginal plug was checked for pregnancy. As the presence or absence of a vaginal plug does not guarantee pregnancy, all rats were kept for three weeks for more certainty and afterward, nonpregnant rats were delivered to the animal house of Hamadan University of Medical Sciences for further use.

## Exercise test

The maximum running capacity was performed individually for maternal (before and after six weeks HIIT period) and pup groups (at 10 weeks old) on the treadmill. First, the warm-up initiated with 5 mins at speed of 5m/min and then each rat started exercise test by 3 mins at speed of 8m/min with 0˚ inclines. The speed of running gradually increased by 3m/min every 3 mins, until signs of exhaustion. Those who refused to run or have uncoordinated steps are determining factors in the detection of exhaustion [18]. Maximum speed and total distance were recorded and used to determine the appropriate speed in HIIT sessions in maternal groups.

## High-intensity-interval training

Training protocol started immediately after 5 mins warm-up and High-intensity-interval-training consist to running on treadmill with the speed of 18m/min at 10˚ inclination for 3 mins (85–95% of VO2max), switching with active recovery, and the speed of 13m/min (65% of VO2max) at 0˚ inclination for 5 days/week which is in accordance with the overload principle [19]; the duration and number of bouts were increased every week. The control group was kept in the training room during the training session to resemble the exercise groups. The HIIT protocol is depicted in Table 1.

## Sirt6 and IGF-2 gene expression by RT-PCR method

The total RNA was extracted from the left ventricle heart using RNA isolation of Kiazol protocol (Kiazist Life Sciences, Iran) and the quantity and quality of extracted RNA were evaluated by Nano-Drop and gel electrophorus. Following, the HyperScript™ synthesis kit (GeneAll, Korea) was utilized for synthesizing the reaction of the first-strand cDNA from total RNA according to the manufacturer's order. Then, the resulting cDNA was applied to Real-time quantitative polymerase chain reaction (PCR) using RealQ Plus 2X Master Mix Green

**Table 1. Characteristic of HIIT protocol prior to and during pregnancy.**

| HIIT protocol prior to pregnancy | |
|---|---|
| First week | Speed: 18 (m/min), 10 bouts |
| Second week | Speed: 18 (m/min), 11 bouts |
| Third week | Speed: 20 (m/min), 12 bouts |
| Fourth week | Speed: 22 (m/min), 13 bouts |
| Fifth week | Speed: 24 (m/min), 14 bouts |
| Sixth week | Speed: 26 (m/min), 15 bouts |
| HIIT protocol during pregnancy | |
| First week | Speed: 18 (m/min), 10 bouts |
| Second week | Speed: 18 (m/min), 11 bouts |
| Third week | Speed: 20 (m/min), 12 bouts |

**Table 2. The PCR program.**

| Cycles | Duration of cycle | Temperature |
|---|---|---|
| 1˚ | 15 minutes | 95˚ |
| 40 | 15 seconds | 95˚ |
| | 30 seconds | 60˚ |
| | 30 seconds | 72˚ |

(AMPLIQON, Denmark) in a three-step PCR program and a total 20 reaction volumes were performed on Real-Time PCR system (Roche Life Sciences, Germany). Relative expression of examined genes was quantified and normalized by b-actin as the internal control gene. Also, conditions used of PCR and upstream and downstream primers were respectively described in Tables 2 and 3.

## Lipid serum by the colorimetric method

Fasting blood samples of offspring were collected from inferior *vena cava* at 10 weeks old and it was centrifuged at 600 g for 10 min. serum high-density lipoprotein (HDL), low-density lipoprotein (LDL), cholesterol (Cho) and triglycerides (TG) were measured using Pars Azmon kit (Tehran, Iran) and Alpha-classic autoanalyzer.

## Statistical analysis

The distribution of data was normal when examined using the Kolmogorov-Simonov test. All data were presented as mean ± SEM. Data were analyzed using GraphPad Prism® 6.0 (Graph-Pad, La Jolla, CA, USA). One-way and Two-way ANOVA with Dunnett's posthoc test was used for statistical analysis. A probability of 0.05 was considered as the criterion for significance.

## Results

### The outcome of maternal characteristics

The maternal characteristics are shown in Table 4. There was no significant difference in maternal weight at the start of work. Our result showed that maternal weight gain was not affected by HIIT prior to or during pregnancy and we did not find a significant difference among the pregnant rats in body weight (P = 0.29). Also, we assessed the exercise test before and after six weeks of HIIT to ensure the effectiveness of HIIT on mothers that are characterized by enhanced distance and speed on the treadmill. As expected, distance (P≤0.0001, F (3,28) = 143.3) and speed (P≤0.0001, F(3,28) = 148.3) of exercised mothers were significantly greater than non-exercised mothers.

**Table 3. Conditions and sequence of primers.**

| Gene | | Sequence | PCR PRODUCT SIZE | AMPLICON $T_m$ |
|---|---|---|---|---|
| Sirt6 | Forward | GAC CTA ACG CTC GCT GAT GA | 163 | 60˚C |
| | Reverse | CCT GGC GGT CAT GTT TTG TG | | |
| IGF-2 | Forward | GAG GGG AGC TTG TTG ACA C | 155 | 60˚C |
| | Reverse | GGC ACA GTA TGT CTC CAG G | | |
| b-actin | Forward | ATC AGC AAG CAG GAG TAC GAT | 94 | 60˚C |
| | Reverse | AAA GGG TGT AAA ACG CAG CTC | | |

**Table 4. Maternal characteristics.**

|  | c | BP | BDP | DP |
|---|---|---|---|---|
| Maternal' weight (gr) |  |  |  |  |
| before 6 weeks | 194.54 ± 3.2 | 193.17 ± 2.3 | 192.25 ± 3.3 | 194.12 ± 3.31 |
| after 6 weeks | 212.87 ± 4 | 208.81 ± 4.77 | 207.12 ± 4.67 | 211.1 ± 4.43 |
| The distance of maternal Exercise test (m) |  |  |  |  |
| before 6 weeks HIIT | 131.50 | 141.4 | 139.4 | - |
| after 6 weeks HIIT | 166.3 | 877.9**** | 861.4**** | - |
| The speed of maternal Exercise test (m/min) |  |  |  |  |
| before 6 weeks HIIT | 15.88 | 17.50 | 16.63 | - |
| after 6 weeks HIIT | 17.75 | 40.63**** | 40.25**** | - |
| Number of pregnant rats/total rats | 5/8 | 6/8 | 4/8 | 6/8 |
| Number of pups | 7 ± 0.92 | 5 ± 1.71 | 9 ± 0.98 | 8 ± 0.61 |
| Birthweight of pups (gr) | 6.435 ± 0.17 | 6.020 ± 0.25 | 6.325 ± 0.20 | 6.045 ± 0.13 |
| Sex distribution |  |  |  |  |
| Female | 4.60 ± 0.50 | 2.83 ± 1.07 | 5.25 ± 0.47 | 4.5 ± 0.76 |
| Male | 3.80 ± 0.58 | 3.16 ± 0.70 | 4.50 ± 0.64 | 5.33 ± 0.49 |

Values represented as the mean ± standard error. Abbreviations: **C group**, sedentary maternal (control); **BP group,** maternal that exercised before pregnancy; **BDP group**, maternal that exercised before and during pregnancy; **DP group**, maternal that exercised during pregnancy.

**** $p < 0.0001$ = significant statistically difference as compared to the Control group. One-way ANOVA followed by Dunnett's multiple comparisons test.

Besides, the number of pups and the sex distribution of pups are shown in Table 4 and the outcomes of the number of pups (P = 0.16) and sex distribution (P = 0.10) were not statistically significant.

## The outcome of offspring

**Maternal HIIT had effect on the weekly weight gain of the offspring whose mothers exercised only during pregnancy.** The offspring were divided into four groups according to the initiation time of maternal exercise. As shown in Table 4 and Fig 2, the initiation time of maternal HIIT does not affect offspring birth weight (P = 0.246); however it could change the weight of offspring at 6[th], 7[th] and 8[th] weeks, as two-way ANOVA revealed a significant interaction between the initiation time of maternal HIIT and weekly weight gain of offspring (p < 0.001, F(21,119) = 8).

Therefore, this data indicated that maternal HIIT only during pregnancy leads to a significant decrease in the weekly weight gain of pups.

**Maternal HIIT prior to and during pregnancy improve running speed and distance of the male offspring.** We compared the offspring exercise test among the pup groups to determine the influence of maternal exercise on the running performance of the male offspring in terms of running speed and tolerance to fatigue. As shown in Fig 3, there was a significant difference among offspring groups. This result indicated that the pups of mothers who exercised prior to and during pregnancy had more running distance (P = 0.0026, F(3,17) = 7.161) and running speed (P = 0.0176, F(3,17) = 4.446) on the treadmill than other pups.

**The initiated period of maternal HIIT has an impact on the heart mass of offspring.** There has not been any difference between the heart mass (P = 0.25) and the heart mass/ body weight ratio (P = 0.31) in pup groups (Table 5).

**Maternal HIIT prior to and during pregnancy have effects on the Sirt6 and IGF-2 gene expression in the heart and serum lipid profile of the offspring.** In order to realize the

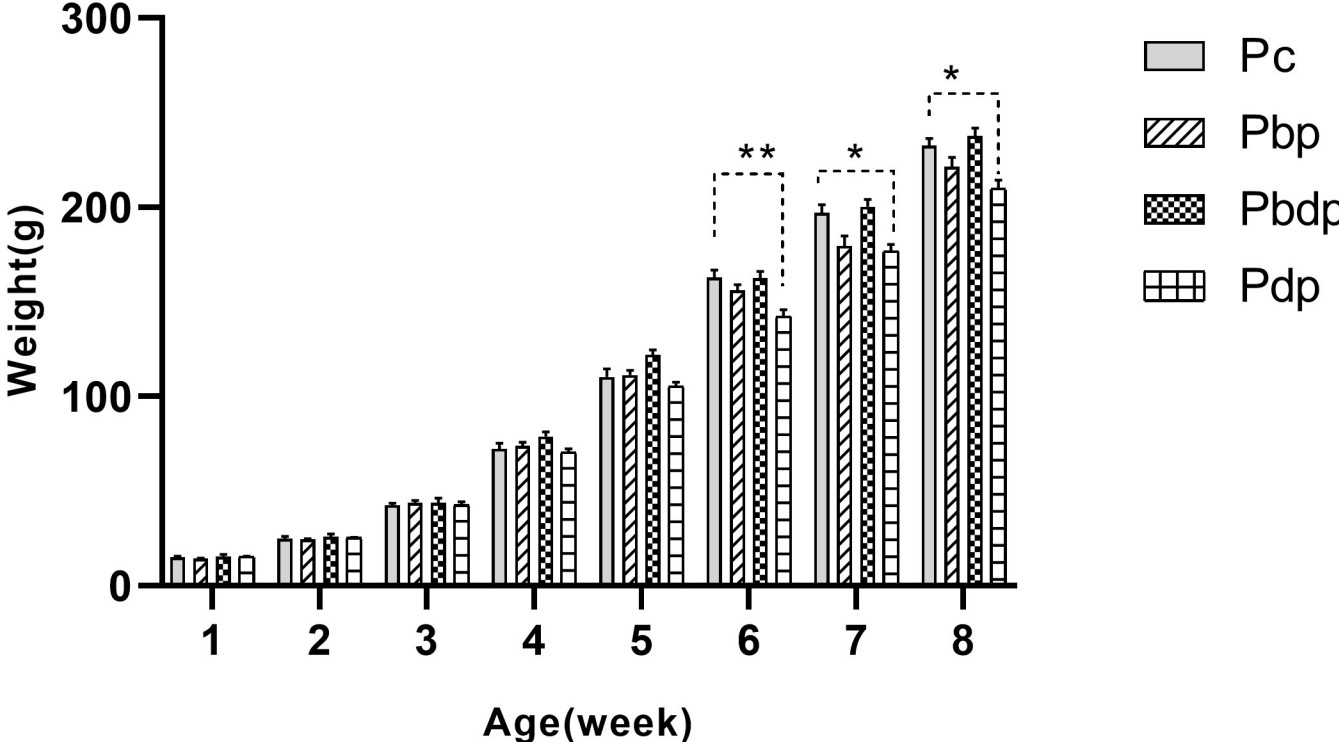

**Fig 2. Comparison of the weekly weight gain among the pup groups.** Abbreviations: **P**c group (n = 5), pups of sedentary mothers; **P**bp group (n = 6), pups of mothers who exercise only before pregnancy; **P**bdp group (n = 4), pups of mothers who exercise prior to and during pregnancy; **P**dp group (n = 6), pups of mothers who exercise only during pregnancy. Data presented as means ± S.E.M. * $p<0.05$ and ** $p<0.01$ vs. Pup Control group. Two-way ANOVA followed by Dunnett's multiple comparisons test.

importance of initiation time of maternal HIIT to confer adult offspring health, we evaluated the cardiac Sirt6 and IGF-2 gene expression and serum lipid profile in pup groups. As shown in Fig 4, there was seen an enhancing effect of maternal HIIT on cardiovascular factors due to a significant increase in Sirt6 gene expression) P = 0.017), which in turn, decreased IGF-2

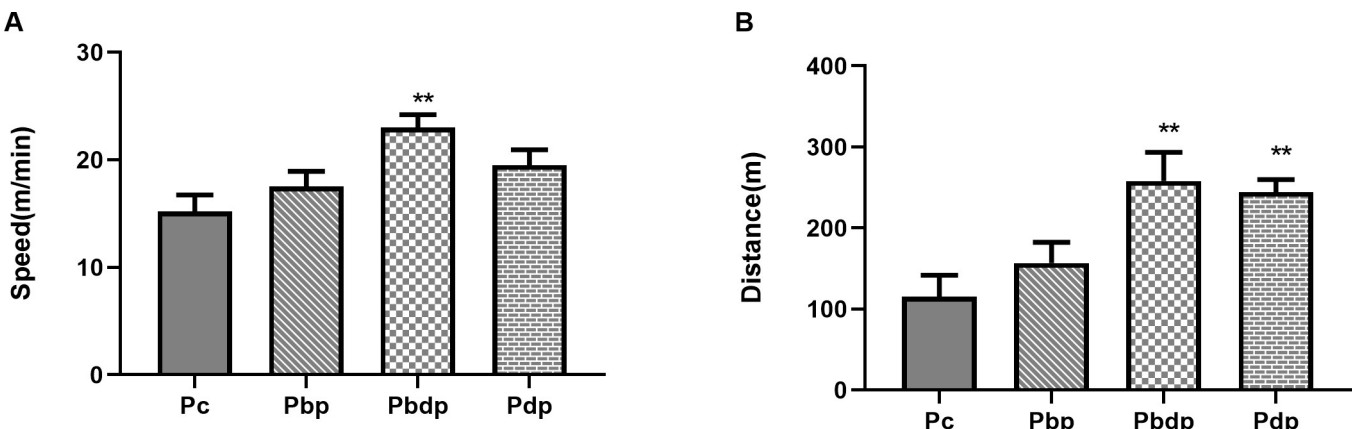

**Fig 3. Comparison of exercise test among the pup groups.** (A) A maximum speed of adult male offspring, (B) A maximum distance of adult male offspring. Abbreviations: **P**c group (n = 5), pups of sedentary mothers; **P**bp group (n = 6), pups of mothers who exercise only before pregnancy; **P**bdp group (n = 4), pups of mothers who exercise prior to and during pregnancy; **P**dp group (n = 6), pups of mothers who exercise only during pregnancy. Data presented as means ± S. E.M. ** $p<0.01$ vs. Pup Control group. One-way ANOVA followed by Dunnett's multiple comparisons test.

**Table 5. Pups weight characteristics.**

| | Pc | Pbp | Pbdp | Pdp |
|---|---|---|---|---|
| | (n = 5) | (n = 6) | (n = 4) | (n = 6) |
| Pup's Heart mass(gr) | 1.136 ± 0.040 | 1.087 ± 0.041 | 1.190 ± 0.034 | 1.083 ± 0.023 |
| Pup's Heart mass/Body weight(gr) percentage | 0.389 ± 0.006 | 0.387 ± 0.008 | 0.393 ± 0.007 | 0.371 ± 0.003 |

Data presented as means ± S.E.M. Abbreviations: **P**c group, pups of sedentary mothers; **P**bp group, pups of mothers who exercise only before pregnancy; **P**bdp group, pups of mothers who exercise prior to and during pregnancy; **P**dp group, pups of mothers who exercise only during pregnancy.

(P = 0.0473) gene expression. The gene expression results of the pup groups suggested that maternal exercise prior to and during pregnancy has beneficial effects on the offspring. Thus, the present study proposes a promising approach in improving the factors involved in the cardiac health of adult offspring as a result of the maternal HIIT.

Evaluation of the lipid profile of male pups showed that maternal exercise prior to and during pregnancy and only during pregnancy has a positive impact on the lipids profile of adult offspring by decreasing serum levels of LDL and Cho.

As shown in Fig 5, the comparison of HDL and TG among the offspring groups show no significant differences while the comparison of LDL (P = 0.0029) and Cho (P = 0.0074) revealed significant differences between **P**c, **P**bdp and **P**dp groups.

## Discussion

Maternal exercise during pregnancy as a positive maternal behavior has been extensively investigated [20–23]. The present study aimed to investigate the effect of physical activity of mothers as a healthy lifestyle on the health of offspring. Our finding indicated that firstly, strenuous intensity could be done in pregnancy without detrimental effect on the birth weight of offspring, and secondly, the ideal time for initiating high-intensity-interval training is prior to and during pregnancy that leads to the increase of Sirt6 gene expression and the decrease of IGF-2 gene expression in the heart, and the decrease LDL and Cho levels in the serum of offspring.

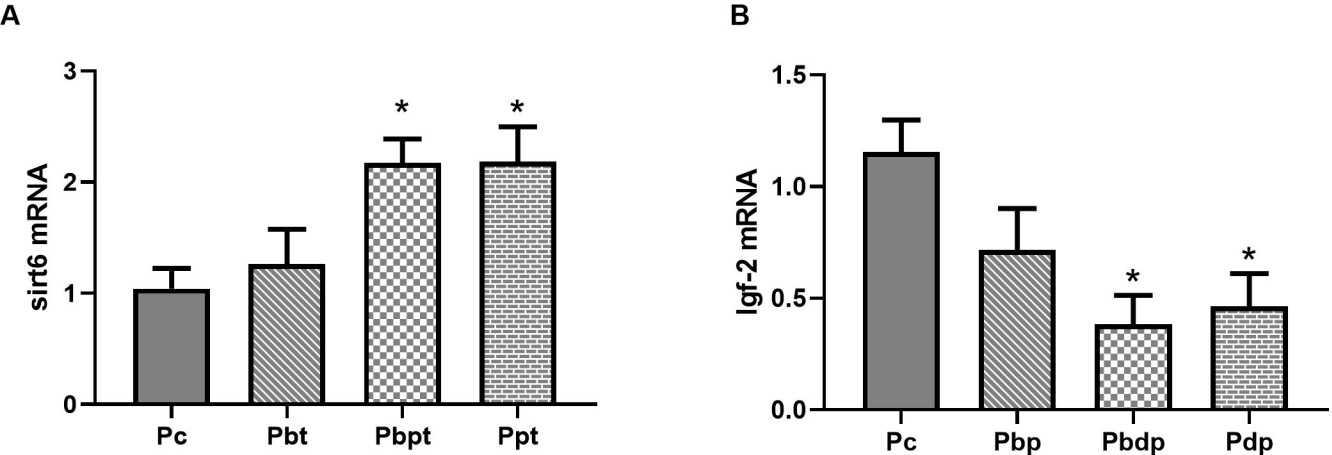

**Fig 4. Comparison of cardiac gene expression among the pup groups.** (A) cardiac Sirt6 expression in adult male pup groups. (B) cardiac IGF-2 expression in adult male pup groups. Abbreviations: **P**c group (n = 5), pups of sedentary mothers; **P**bp group (n = 6), pups of mothers who exercise only before pregnancy; **P**bdp group (n = 4), pups of mothers who exercise prior to and during pregnancy; **P**dp group (n = 6), pups of mothers who exercise only during pregnancy. Data presented as means ± S.E.M. * p<0.05 vs. Pup Control group. One-way ANOVA followed by Dunnett's multiple comparisons test.

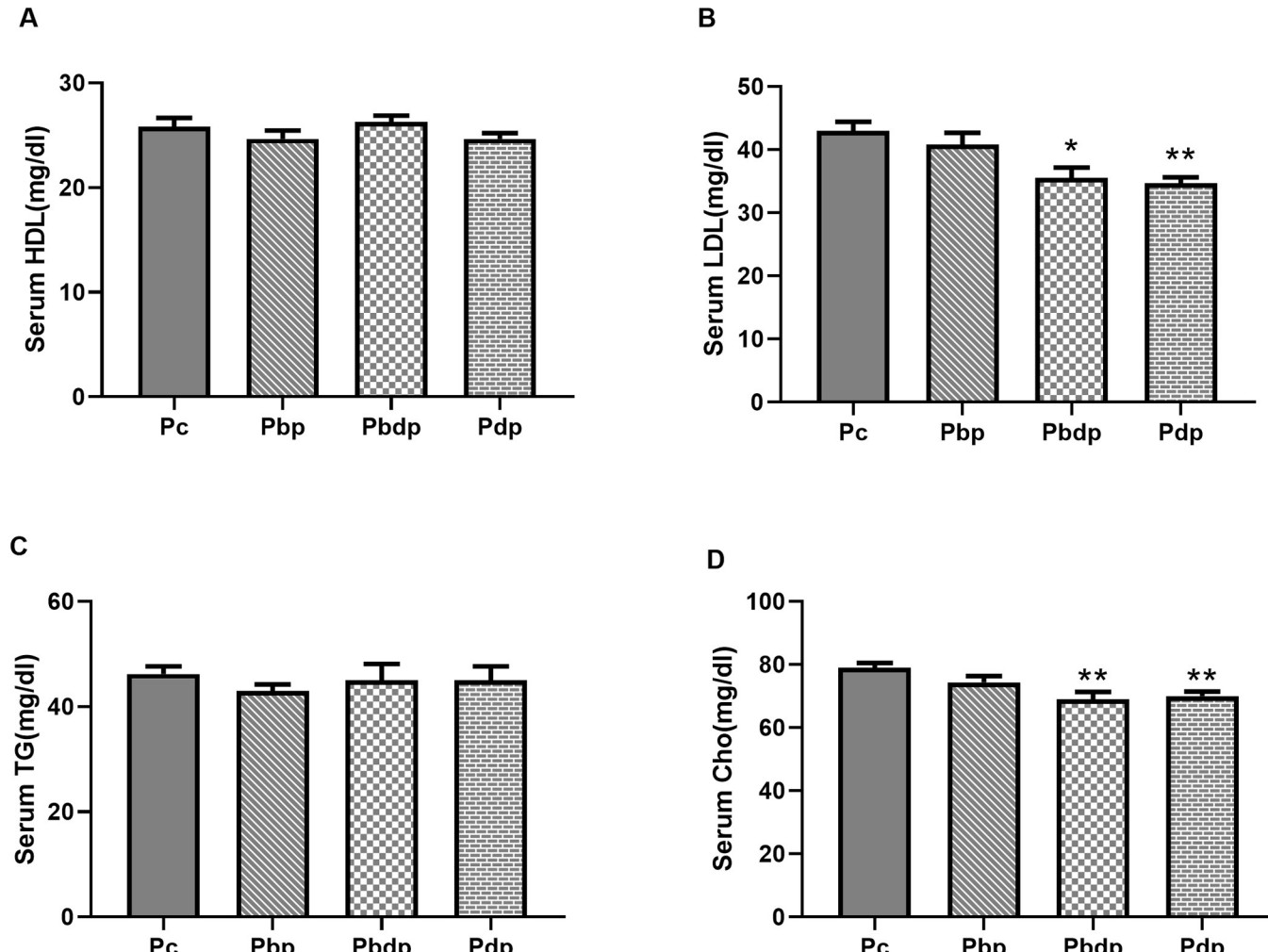

**Fig 5. The comparison of lipid profile among the pup groups.** (A) The serum level of HDL in male adult offspring groups. (B) the serum level of LDL in male adult offspring. (C) The serum level of TG in male adult offspring groups. (D) The serum level of Cho in male adult offspring groups. Abbreviations: **P**c group (n = 5), pups of sedentary mothers; **P**bp group (n = 6), pups of mothers who exercise only before pregnancy; **P**bdp group (n = 4), pups of mothers who exercise prior to and during pregnancy; **P**dp group (n = 6), pups of mothers who exercise only during pregnancy. Data presented as means ± S.E.M. * p<0.05, ** p<0.01 vs. Pup Control group. One-way ANOVA followed by Dunnett's multiple comparisons test.

To date, only one other study evaluated HIIT during pregnancy which is consistent with our observation [13]. Our study indicates that pregnant rats are capable to perform HIIT on the treadmill throughout pregnancy. To our knowledge, the present study is the first study that compares the effects of preconception and prenatal periods of maternal HIIT or directly both of them on the weight and cardiovascular health of offspring. Scientific evidence supports a relationship between birth weight and postnatal health outcomes [24] and mentions the birth weight as an important factor in adult disease [25, 26]. Consistent with the previous literature [27], our study found that maternal exercise did not affect the birth weight of offspring. The weighted result of animals was notable in that the birth weight of pups was not affected by the initiation period of maternal HIIT, but maternal HIIT has a significant effect on the weekly weight gain of pups at 6th-8th weeks. Also, the result of heart weight and heart weight/ body weight ratio of offspring indicated that maternal HIIT only during pregnancy could lead to a significant decrease as compared to the pup of mothers who exercised prior to and during

pregnancy. Our observations reveal that high-intensity exercise only during pregnancy leads to the reduction of the body weight of pups at 6th -8th weeks, while this weight loss was not observed in pups of mothers who were compatible with this intensity before pregnancy. This decrease may be due to a stress effect of exposing the mother to exercise, commencing only in pregnancy - which induces a stress response (unlike the Pbdp group, whose mothers were exercised prior to pregnancy as well).

Thus, the current result highlighted the importance of the preconception period for maternal physiological adaptation to high-intensity-interval training before pregnancy. It is noteworthy that the observed decrease in the adult weight of pups does not mean a negative impact and needs to be evaluated with anthropometric indicators, cardiac function and histologic examination in future works.

The epidemiological observations highlighted that the developmental changes of the pups related to health were associated with the prenatal environment. Given that HIIT could be tolerated by pregnant rats without an effect on the birth weight of pups, the main question is whether strenuous exercise could modify the intrauterine environment to provide rich conditions concerning the cardiovascular factors which affect the developmental status. Voluntary maternal exercise on the first day of pregnancy could modulate mitochondrial gene expression of the fetal heart [28], although it is not clear if elevated gene expression would persist in adulthood. Another study suggested a molecular positive change in the hippocampus of rats by controlled maternal exercise during pregnancy [29]. Raipura et al. have shown that voluntary exercise during pregnancy seems to decrease the metabolic risk in offspring [27]. The previous studies on humans and animals have shown a positive effect of maternal exercise on offspring; however, the optimal timing of initiated exercise to confer cardiovascular benefits to the next generation has not been addressed. The findings of Brito et al. demonstrated the protective effects of moderate maternal exercise in the hearts of progeny by modulating the oxidative stress and Sirtuin6 protein levels [5]. One of the well-known pathways of exercise for cardioprotective effects is the activation of the sirtuin family [9, 30]. There is evidence that down-regulation IGF-AKT signaling directly by Sirt6 is capable of blocking heart failure and myocardial disease [10]. Also, studies on the fetal left ventricle have shown that an elevation of IGF-2 gene expression levels and its receptor can increase the risk of cardiovascular disease in adulthood [31]. Darby et al. have suggested that maternal undernutrition has negative consequences for cardiac health due to an up-regulation of IGF-2 signaling in the fetal heart [32]. Therefore, the present study assesses the possibility of activation of the cardiac gene expression related to cardioprotection through the maternal exercise in adult offspring. Consistent with our hypothesis, it is indicated that maternal HIIT both prior to and during pregnancy or only during pregnancy compared with maternal exercise before pregnancy could increase Sirt6 gene expression and decrease IGF-2 gene expression in the hearts of the offspring. Thus, it was concluded that maternal HIIT during pregnancy helps giving birth to offspring with enhanced conditions in regard to the factors involved in the heart in later life.

Additionally, given that the cardiovascular adaptations to strenuous exercise related to health are due, in part, to the regulation of the level of serum lipids [33–35], the offspring's serum lipid profile as the index of cardiovascular health was evaluated. Maternal voluntary exercise prior to and during pregnancy was found to improve glucose homeostasis in adult offspring [36]. A similar study has shown that, in C57BL/6 mice, maternal exercise before and during gestation has effects on offspring metabolic health without changes in body composition [36]. A study of rabbits demonstrated that cholesterol-lowering interventions during pregnancy have profound effects on the vascular health of adult offspring [37]. The obtained data show that maternal HIIT prior to and during pregnancy could decrease LDL and Cho levels of offspring, but do not have a significant effect on serum HDL and TG levels. It is noted

that maternal exercise prior to and during pregnancy is the same, as only during pregnancy could alter the lipid profile of offspring (LDL and Cho). Thus, the present study indicates that maternal HIIT as a suitable intervention to reduce the lipid profile has long-term beneficial effects in adult male offspring.

Moreover, as expected, the offspring of mothers who exercised prior to and during pregnancy can attain higher speed and greater distance in comparison to other pups. Since the maximal oxygen uptake ($VO_2max$) could be predicted by the data of the submaximal exercise test [38, 39], the results of the exercise test of offspring could indirectly indicate the effect of maternal exercise before and during pregnancy on the increase of offspring's physical readiness through the increased running speed and distance. Offspring's exercise test data which was in accordance with the previous study [23] showed that the mother's activity prior to and during pregnancy increased running speed and time to fatigue of offspring.

To summarize, the present study reinforces the beneficial effects of maternal HIIT prior to and during pregnancy as optimal periods of promoting health of offspring by changes in serum lipid profile, cardiac gene expression and running performance. Therefore, based on the results of present study maternal exercise could be suggested as an intervention to improve the health and physical readiness of male offspring.

## Supporting information

**S1 Fig. Amplification plots and melting cure of the selected genes.** A) melting cure of sirt6, B) melting cure of igf2, C) melting cure of b-actin, D) Amplification plots of sirt6, E) Amplification plots of igf2 and F) Amplification plots of b-actin.
(TIF)

**S1 File. Minimal data set of the pup's exercise test.** Abbreviations: **P**c group (n = 5), pups of sedentary mothers; **P**bp group (n = 6), pups of mothers who exercise only before pregnancy; **P**bdp group (n = 4), pups of mothers who exercise prior to and during pregnancy; **P**dp group (n = 6), pups of mothers who exercise only during pregnancy.
(XLSX)

**S2 File. Minimal data set of the pup's heart weight.** Abbreviations: **P**c group (n = 5), pups of sedentary mothers; **P**bp group (n = 6), pups of mothers who exercise only before pregnancy; **P**bdp group (n = 4), pups of mothers who exercise prior to and during pregnancy; **P**dp group (n = 6), pups of mothers who exercise only during pregnancy.
(XLSX)

**S3 File. Minimal data set of the pup's weekly weight gain.** Abbreviations: **P**c group (n = 5), pups of sedentary mothers; **P**bp group (n = 6), pups of mothers who exercise only before pregnancy; **P**bdp group (n = 4), pups of mothers who exercise prior to and during pregnancy; **P**dp group (n = 6), pups of mothers who exercise only during pregnancy.
(XLSX)

**S4 File. Minimal data set of the pup's gene expression.** Abbreviations: **P**c group (n = 5), pups of sedentary mothers; **P**bp group (n = 6), pups of mothers who exercise only before pregnancy; **P**bdp group (n = 4), pups of mothers who exercise prior to and during pregnancy; **P**dp group (n = 6), pups of mothers who exercise only during pregnancy.
(XLSX)

**S5 File. Minimal data set of the pup's lipid profile.** Abbreviations: **P**c group (n = 5), pups of sedentary mothers; **P**bp group (n = 6), pups of mothers who exercise only before pregnancy; **P**bdp group (n = 4), pups of mothers who exercise prior to and during pregnancy; **P**dp group

(n = 6), pups of mothers who exercise only during pregnancy.
(XLSX)

## Acknowledgments

The authors would like to thank the Neurophysiology Research Center because of helping this project and Dr. Seyed Asaad Karimi for the reading of the manuscript.

## Author Contributions

**Conceptualization:** Iraj Salehi.

**Data curation:** Reihaneh Mohammadkhani, Hamid Rajabi.

**Formal analysis:** Reihaneh Mohammadkhani, Alireza Komaki.

**Funding acquisition:** Iraj Salehi, Alireza Komaki.

**Investigation:** Reihaneh Mohammadkhani.

**Methodology:** Reihaneh Mohammadkhani, Neda Khaledi, Alireza Komaki.

**Project administration:** Neda Khaledi, Iraj Salehi, Alireza Komaki.

**Software:** Reihaneh Mohammadkhani.

**Supervision:** Neda Khaledi, Hamid Rajabi, Iraj Salehi.

**Writing – original draft:** Reihaneh Mohammadkhani.

**Writing – review & editing:** Neda Khaledi, Iraj Salehi, Alireza Komaki.

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
