## [Decision Letter · Decision Letter 0]

4 Feb 2020

PONE-D-19-33225

Influence of maternal high-intensity-interval training prior to and during pregnancy on the cardiovascular health of the male offspring

PLOS ONE

Dear Dr. khaledi,

Thank you for submitting your manuscript to PLOS ONE. After careful consideration, we feel that it has merit but does not fully meet PLOS ONE’s publication criteria as it currently stands. Therefore, we invite you to submit a revised version of the manuscript that addresses the points raised during the review process.

There are some concerns about the methods or reporting of the methods. Equally the conclusions are far to much to be justified on the data provided 

We would appreciate receiving your revised manuscript by Mar 20 2020 11:59PM. To enhance the reproducibility of your results, we recommend that if applicable you deposit your laboratory protocols in protocols.io, where a protocol can be assigned its own identifier (DOI) such that it can be cited independently in the future. For instructions see: http://journals.plos.org/plosone/s/submission-guidelines#loc-laboratory-protocols

We look forward to receiving your revised manuscript.

Kind regards,

Christopher Torrens

Academic Editor

PLOS ONE

Journal Requirements:

2. To comply with PLOS ONE submissions requirements, please provide methods of sacrifice (for both mothers and offspring, including the ones mentioned at line 108) in the Methods section of your manuscript.

3. Please consider modifying your title to ensure that it is specific, descriptive and  concise, by specifying that the study was performed in rats.

4. We suggest you thoroughly copyedit your manuscript for language usage, spelling, and grammar. If you do not know anyone who can help you do this, you may wish to consider employing a professional scientific editing service.  

Additional Editor Comments (if provided):

Mating: The methods state that 32 Wistar rats were purchased, giving 8 per maternal exercise group. Later in the mating section it states that non-pregnant rats were discarded. So, either more than 32 rats were obtained or no rats were discarded for being non-pregnant making the statement redundant. Please clarify.

Line 107: "...some pups were eliminated...". Please give the method of this (e.g. cervical dislocation). Also if litters were standardised, what were they standardised to?

Line 130: "those that refused to run...". How many refused to run, per group per protocol?

line 153: "...normalised by b-actin." It is preferable to use multiple housekeeping reference genes to account for treatment-induced changes in reference gene expression. Since there is only one here, do you know is b-actin expression is stable across exposures of maternal exercise?

Line 199: Two-way ANOVA is not mentioned in the methods but is mentioned here

Reviewers' comments:

Reviewer's Responses to Questions

**Comments to the Author**

1. Is the manuscript technically sound, and do the data support the conclusions?

Reviewer #1: No

Reviewer #2: Partly

2. Has the statistical analysis been performed appropriately and rigorously? 

Reviewer #1: Yes

Reviewer #2: I Don't Know

3. Have the authors made all data underlying the findings in their manuscript fully available?

Reviewer #1: Yes

Reviewer #2: Yes

4. Is the manuscript presented in an intelligible fashion and written in standard English?

Reviewer #1: Yes

Reviewer #2: Yes

5. Review Comments to the Author

Reviewer #1: Mohammadkhani et al. present an interesting manuscript investigating the effect of maternal HIIT on cardiovascular health of male offspring.

The authors conclude that maternal exercise reduces the risk of heart disease and may have an intergenerational effect – there is no data to support these claims.

Studies showing that maternal exercise increases Sirt6 and Igf2 in the heart have already been published.

The authors present data showing that heart mass is different in offspring from dams that exercised only during pregnancy – what does this mean?

The authors state that, “The results of the gene expression of groups of pups suggested that maternal exercise only prior to pregnancy or prior to and during pregnancy has a cardioprotective effect on the next generation. Thus, the present study highlights the pregnancy period as a vital period with a long-lasting effect on the cardiac health of offspring.” This is a huge overstatement. There is no way to conclude this based on expression of 2 genes. Further, it is not stated at what age gene expression is measured in the offspring.

It is also not clear what age the mice were when LDL/HDL/etc. was measured.

Reviewer #2: The goal of the study by Reihaneh et al was to determine the effects of maternal high-intensity-interval training (HIIT) on cardiovascular health of male offspring. The authors found that maternal HITT before and during pregnancy increased exercise capacity and decreased serum levels of LDL and cholesterol in male offspring at 10 weeks of age. Maternal training also improved two cardioprotective factors in offspring hearts, increasing mRNA expression of Sirt6 and decreasing mRNA expression of Igf2. The topic and data are interesting and have potentially important clinical relevance. However, the very limited data are insufficient to prove that maternal exercise results in cardiovascular protection in offspring. To prove this the authors would need to perform many additional experiments including direct measurements of cardiac function. The manuscript would be much stronger if there were also data on Sirt family gene expression, cardiac and skeletal muscle histology, and more detailed serum analysis. In addition, the lack of n for each figure legend and table is concerning.

1. In addition to function studies, does maternal HIIT affect the histology of hearts? Longitudinal sections of whole mouse hearts and microscopic views of the inter-ventricular septa and left ventricles may suggest the improvement of cardiovascular function in offspring heart.

2. Sirt1, Sirt2, Sirt4, Sirt5, and Sirt6 are highly expressed in heart and known to be involved in cardiovascular disease protection. Does maternal HIIT change these gene expression? Also, the expression of Sirt6 targets (Nrf2 and NFkappaB) and the change of histone acetylation (ex. H3K9ac, H3K18ac) will support the importance of Sirt6 in the effects of maternal HIIT on offspring heart.

3. What is the n for each experiment? It is very odd that this is not included in each figure legends. Table 4 on Maternal Characteristics is not informative in terms of understanding the n for studies of offspring groups. Is the n for all of the offspring experiments the number of pregnant rats? The n for studies of offspring is based on the number of mothers studied, not the number of offspring. Thus, if for example in the BP group there are 6 mothers that each have a litter of 3 males, the n=6. If this information is not provided, the data cannot be evaluated.

-

4. On page 5 it is stated that you eliminated some pups to create uniform conditions. Explain this uniform condition. How many pups were kept per litter? Did you study all pups or pooled?

5. Why were the offspring only studied until 8-10 weeks of age? This is considered young adulthood, and at this age there is less chance of developing cardiovascular disease.

6. Is serum level of IGF2 changed in offspring?

7. Does maternal HIIT affect the expression of Insulin receptor, IGF1 receptor, and IGF2 receptor in offspring heart? Are total and phosphorylated levels of Akt changed by maternal HIIT in offspring heart?

8. Can the authors explain the increase of exercise capacity in offspring by a skeletal muscle phenotype? For example, endurance strength, muscle weight, and fiber composition (cross-sectional area and fiber distribution in muscle).

9. There is no rationale for only studying male offspring.

10. Page 8. Fasting blood samples were collected; how long were the animals fasted?

11. Wistar rats, not Wister (page 4 line 81).

12. Page 12. The gene expression data in the text do not seem correct (line 239).

6. PLOS authors have the option to publish the peer review history of their article (what does this mean?). If published, this will include your full peer review and any attached files.

Reviewer #1: No

Reviewer #2: No

---

## [Author Response · Author response to Decision Letter 0]

17 Apr 2020

Dear Christopher Torrens

Academic Editor

PLOS ONE

The authors would like to Thank you for giving me the opportunity to submit a revised draft of our manuscript titled [Influence of the maternal high-intensity-interval-training on the cardiac Sirt6 and lipid profile of the adult male offspring in rats, No. PONE-D-19-33225] to [PLOS ONE]. We appreciate the time and effort that you and the reviewers have dedicated to providing your feedback on our manuscript. We have been able to incorporate changes to reflect most of the suggestions provided by the reviewers. We have highlighted the changes within the manuscript. Here is a point-by-point response to the comments and concerns. We hope that this new version of the manuscript is satisfactory for the presentation and we are thankful once again to the reviewers for their useful comments.

Sincerely yours

Neda Khaledi (Ph. D)

Corresponding Author

 

Academic editor' comments:

We truly appreciate the Editor’s suggestion, and we recognize the benefits improving the quality of the manuscript. 

 Point 1: Please ensure that your manuscript meets PLOS ONE's style requirements, including those for file naming.

Response: The manuscript was modified based on PLOS ONE's style.

Point 2: To comply with PLOS ONE submissions requirements, please provide methods of sacrifice (for both mothers and offspring, including the ones mentioned at line 108) in the Methods section of your manuscript.

Response: We have added the method of sacrifice in Materials and Methods section. Following the animal care guidelines of the university, some were delivered to the animal house of Hamadan University of Medical Sciences; however, on the other hand some other which could still provide further data were used in other investigations. while some other that could still provide further data was used in other investigations. 

Point 3: Please consider modifying your title to ensure that it is specific, descriptive and concise, by specifying that the study was performed in rats.

Response: In accordance with your suggestion, the title has been changed and it is now “Influence of the initiated period of maternal high-intensity-interval-training on the profile lipid and cardiac sirt6 of the adult male offspring in rats”.

Point 4: We suggest you thoroughly copyedit your manuscript for language usage, spelling, and grammar. If you do not know anyone who can help you do this, you may wish to consider employing a professional scientific editing service.

Response: By getting the help of a native English-speaking associate we improved our manuscript.

Point 5: Mating: The methods state that 32 Wistar rats were purchased, giving 8 per maternal exercise group. Later in the mating section it states that non-pregnant rats were discarded. So, either more than 32 rats were obtained or no rats were discarded for being non-pregnant making the statement redundant. Please clarify.

Response Apologies if the part was not clear enough in the original manuscript. 32 female Wistar rats were purchased. Animals were randomly divided into two maternal groups; maternal sedentary (control group, n=8) and maternal exercise groups (n=24). Maternal exercise groups were also divided into 3 subgroups (BP, BDP and DP groups) based on the initiation period of the maternal exercise. Mothers that were not pregnant, at the end of three weeks pregnancy period, were eliminated from the study. As a result, five pregnant rats in the control group, six pregnant rats in the BP group, four pregnant rats in the BDP group and six pregnant rats in the DP group remained in the study with their five offspring (to create uniform condition). At first, we randomly select two male pups from each pregnant rat per group but in accordance with the suggestion of reviewer 2, n for offspring group were modified based on the pregnant rat.

Point 6: Line 107: "...some pups were eliminated...". Please give the method of this (e.g. cervical dislocation). Also if litters were standardised, what were they standardised to?

Response: following the animal care guidelines of the university, some were delivered to the animal house of Hamadan University of Medical Sciences; however, on the other hand some other which could still provide further data were used in other investigations. while some other that could still provide further data was used in other investigations. Also, the litter size was standardized based on the lowest litter size (the BP group: 2 female and 3 male) on the first day to create uniform conditions for postnatal growth and development of the pups. 

Point 7: Line 130: "those that refused to run...". How many refused to run, per group per protocol?

Response: These conditions were observed during the exercise test (not during the main protocol of exercise. An exercise test is the one that is performed by progressively increasing workloads up to limiting fatigue caused by exhaustion. Finally, all rats refused to run in the exercise test and the maximum speed obtained from this test were used as an indicator of speed in the main protocol exercise. 

Point 8: line 153: "...normalised by b-actin." It is preferable to use multiple housekeeping reference genes to account for treatment-induced changes in reference gene expression. Since there is only one here, do you know is b-actin expression is stable across exposures of maternal exercise?

Response: The b-actin gene has been used as a reference for maternal exercise in several studies resembling our work [1-4].

Point 9: Line 199: Two-way ANOVA is not mentioned in the methods but is mentioned here.

Response: It was corrected.

Reviewers' comments:

Reviewer #1: Manuscript Number: PONE-D-19-33225

We are grateful to the reviewer 1 for their insightful comments on this article. We have been able to incorporate changes to reflect most of the suggestions provided by the reviewer.

Reviewer #1: Mohammadkhani et al. present an interesting manuscript investigating the effect of maternal HIIT on the cardiovascular health of male offspring.

Response: Thank you for the positive comment. 

Point 1: The authors conclude that maternal exercise reduces the risk of heart disease and may have an intergenerational effect – there is no data to support these claims.

Response: You have raised an important point here. It is well known that calorie restriction and physical exercise can effectively modulate the activity of Sirtuins towards well-being and cardiovascular health [5, 6]. We used the results of Sundaresan et al. which indicated an antihypertrophic effect of sirt6 in cardiomyocytes and showed a low level of sirt6 be a risk factor for heart failure concluded that maternal exercise with increase sirt6 and decrease igf2 genes reduces the risk of heart disease [7]. Also, others studies showed that SIRT6 has a major role in regulating the expression of IGF signaling–related genes and the dysregulation of this pathway might contribute to the pathophysiology of several diseases, including heart failure [4, 8, 9]. However, in accordance with your suggestion, the conclusion has been revised.

Point 2: Studies showing that maternal exercise increases Sirt6 and Igf2 in the heart have already been published.

Response: The effects of maternal exercise in offspring heart are not well understood. Chung et al. demonstrated that maternal voluntary exercise initiated at day 1 of gestation could transfer the positive mitochondrial phenotype to fetal hearts [1]. Also, Songstand et al. showed that the genes related to oxidative stress were altered by maternal HIIT training in the fetal heart [10]. Britro et al concluded intergenerational protective effects of maternal treadmill exercise by increasing the sirt6 protein levels in the hearts of progeny [11]. Most research evaluated the effects of voluntary wheel running or treadmill training with moderate intensity on the heart of fetal. However, the effect of maternal high-intensity interval training on the lipid profile and cardioprotective genes (sirt6 & igf2) of adult male offspring have not been investigated. 

Point 3: The authors present data showing that heart mass is different in offspring from dams that exercised only during pregnancy – what does this mean?

Response: Given that heart mass results from the complex interaction between genetic, environmental, and lifestyle factors, the studies indicated high-intensity interval training could be an effective exercise program for improving cardiac function [12] and could change the heart mass [13, 14]. So, we sought to know whether maternal high-intensity interval training leads to changes in the heart mass of offspring. Thus, it was found that the initiated period of maternal HIIT is an important factor in the heart mass of offspring. Further studies are required to elucidate the echocardiographic parameters offspring underlying the maternal high-intensity interval training correlations for heart mass.

Point 4: The authors state that, "The results of the gene expression of groups of pups suggested that maternal exercise only prior to pregnancy or prior to and during pregnancy has a cardioprotective effect on the next generation. Thus, the present study highlights the pregnancy period as a vital period with a long-lasting effect on the cardiac health of offspring." This is a huge overstatement. There is no way to conclude this based on the expression of 2 genes. Further, it is not stated at what age gene expression is measured in the offspring.

Response: We agree with your suggestion and have revised our conclusion as follows: “The gene expression results of the pup groups suggested that maternal exercise prior to and during pregnancy has beneficial effects on the next generation. Thus, the present study proposes a promising approach in improving the factors involved in cardiac health of the adult offspring as a result of the maternal HIIT”. Also, all pups were sacrificed at 10 weeks old and the left ventricle was extracted. 

Point 5: It is also not clear what age the mice were when LDL/HDL/etc. was measured

Response: Apologies if the part was not clear enough in the original manuscript. All pups were sacrificed at 10 weeks old and the whole of blood was extracted. 

Reviewer #2: The goal of the study by Reihaneh et al was to determine the effects of maternal high-intensity-interval training (HIIT) on cardiovascular health of male offspring. The authors found that maternal HITT before and during pregnancy increased exercise capacity and decreased serum levels of LDL and cholesterol in male offspring at 10 weeks of age. Maternal training also improved two cardioprotective factors in offspring hearts, increasing mRNA expression of Sirt6 and decreasing mRNA expression of Igf2. The topic and data are interesting and have potentially important clinical relevance. However, very limited data are insufficient to prove that maternal exercise results in cardiovascular protection in offspring. To prove this the authors would need to perform many additional experiments including direct measurements of cardiac function. The manuscript would be much stronger if there were also data on Sirt family gene expression, cardiac and skeletal muscle histology, and more detailed serum analysis. In addition, the lack of n for each figure legend and table is concerning.

Response: Thank you for the positive comment. We agree with you that additional experiments including direct measurements of cardiac function are very useful and strengthen our work, but unfortunately due to the lack of facilities and the lack of time we are unable to perform further test at this time. Also, more detailed serum analysis and n for each group were added as advised.

Point 1: In addition to function studies, does maternal HIIT affect the histology of hearts? Longitudinal sections of whole mouse hearts and microscopic views of the inter-ventricular septa and left ventricles may suggest the improvement of cardiovascular function in offspring heart.

Response: Thank you for this suggestion. It would have been interesting to explore this aspect. However, this is beyond the scope of this study, and we will certainly incorporate this in future projects.

Point 2: Sirt1, Sirt2, Sirt4, Sirt5, and Sirt6 are highly expressed in heart and known to be involved in cardiovascular disease protection. Does maternal HIIT change these gene expression? Also, the expression of Sirt6 targets (Nrf2 and NFkappaB) and the change of histone acetylation (ex. H3K9ac, H3K18ac) will support the importance of Sirt6 in the effects of maternal HIIT on offspring heart.

Response: You have raised a valuable point and we agree with your suggestion that the expression of Sirt6 targets will support the importance of our work. We will certainly use your suggestion in future work. However, in cardiovascular diseases, Sirtuins have gained interest in their protective effects. We believe that Sirt6 would be more appropriate than other families because sirt6 appears to have an important role in the heart [15] and cardiovascular disease including cardiac hypertrophy, heart failure and myocardial hypoxic damage [7, 16, 17]. Moreover, physical exercise can effectively modulate the activity of Sirtuins, particularly Sirt6 [15, 18, 19]. The previous experimental study demonstrated that maternal moderate-intensity training during pregnancy could increase the cardiac protein levels of Sirt6 in neonatal of the rat. Thus, we evaluated the effect of maternal strenuous intensity training in the different initiated period on the adult offspring heart. 

Point 3: What is the n for each experiment? It is very odd that this is not included in each figure legends. Table 4 on Maternal Characteristics is not informative in terms of understanding the n for studies of offspring groups. Is the n for all of the offspring experiments the number of pregnant rats? The n for studies of offspring is based on the number of mothers studied, not the number of offspring. Thus, if for example in the BP group there are 6 mothers that each have a litter of 3 males, the n=6. If this information is not provided, the data cannot be evaluated.

Response: You have raised a valuable fact here. In accordance with your point, we changed the number of offspring for each group based on the number of pregnant mothers and analyzed again. The n for each experiment has been added to legend of each table and figure. 

Point 4: On page 5 it is stated that you eliminated some pups to create uniform conditions. Explain this uniform condition. How many pups were kept per litter? Did you study all pups or pooled? 

Response: The litter size was standardized based on the lowest litter size (BP group: 2 female and 3 male) on the first day to create uniform conditions for postnatal growth and development of the pups. All male pups were involved in the study.

Point 5: Why were the offspring only studied until 8-10 weeks of age? This is considered young adulthood, and at this age, there is less chance of developing cardiovascular disease. 

Response: We appreciate the reviewer's comment and are agree with that. Given this, the majority of studies evaluated the effect of maternal exercise in neonatal and demonstrated the levels of Sirt6 expression were increased in the neonatal cardiomyocytes from exercised mothers during pregnancy [11]. We aimed to evaluate at a mature age that many developmental processes are ongoing.

Point 6: Is the serum level of IGF2 changed in offspring? 

Response: In accordance with our purpose IGF2 was measured only in heart tissue.

Point 7: Does maternal HIIT affect the expression of Insulin receptor, IGF1 receptor, and IGF2 receptor in offspring heart? Are total and phosphorylated levels of Akt changed by maternal HIIT in offspring heart?

Response: Thank you for your suggestion. It would have been interesting to explore this aspect. However, in our case, it was not possible due to lack of time. We will certainly measure this in future projects. 

Point 8: Can the authors explain the increase of exercise capacity in offspring by a skeletal muscle phenotype? For example, endurance strength, muscle weight, and fiber composition (cross-sectional area and fiber distribution in muscle).

Response: Given that the maximal exercise test is a useful method for physical capacity, it was demonstrated a strong relationship between treadmill running speed and vo2max in a rat model [20]. Also, the maximum exercise test , a simple methodology and low cost, highlighted as an indicator of exercise capacity to investigate the benefits of exercise training in male rats [21]. So, our study aimed to determine the effect of the initiated period of maternal exercise before and during pregnancy on maximum exercise test of male offspring so the skeletal muscle phenotype was not measured. 

Point 9: There is no rationale for only studying male offspring.

Response: The majority of studies investigating the effects of maternal exercise on offspring have primarily studied the male offspring [22, 23] and Gaini et al demonstrated that improving the mother's physical fitness before and during pregnancy had positive changes in the lipid profile of male offspring. Also, in this study female offspring were used for separate experiments, and only data from male offspring were reported here.

Point 10: Page 8. Fasting blood samples were collected; how long were the animals fasted?

Response: Fasting blood samples were collected after 12 hours of fasting.

Point 11: Wistar rats, not Wister (page 4 line 81).

Response: This mistake was corrected. Thank you.

Point 12: Page 12. The gene expression data in the text do not seem correct (line 239).

Response: Thank you for pointing this out. It was corrected.

1. Chung, E., et al., Maternal exercise upregulates mitochondrial gene expression and increases enzyme activity of fetal mouse hearts. Physiological Reports, 2017. 5(5): p. e13184.

2. Dayi, A., et al., Maternal aerobic exercise during pregnancy can increase spatial learning by affecting leptin expression on offspring's early and late period in life depending on gender. The Scientific World Journal, 2012. 2012.

3. Park, J.-w., et al., Maternal exercise during pregnancy affects mitochondrial enzymatic activity and biogenesis in offspring brain. International Journal of Neuroscience, 2013. 123(4): p. 253-264.

4. Winnik, S., et al., Protective effects of sirtuins in cardiovascular diseases: from bench to bedside. European heart journal, 2015. 36(48): p. 3404-3412.

5. Vitiello, M., et al., Multiple pathways of SIRT6 at the crossroads in the control of longevity, cancer, and cardiovascular diseases. Ageing research reviews, 2017. 35: p. 301-311.

6. Xu, S., P. Bai, and Z.G. Jin, Sirtuins in cardiovascular health and diseases. Trends in endocrinology and metabolism: TEM, 2016. 27(10): p. 677.

7. Sundaresan, N.R., et al., The sirtuin SIRT6 blocks IGF-Akt signaling and development of cardiac hypertrophy by targeting c-Jun. NATURE MEDICINE, 2012. 18(11): p. 1643.

8. Chu, C.-H., et al., Activation of insulin-like growth factor II receptor induces mitochondrial-dependent apoptosis through Gαq and downstream calcineurin signaling in myocardial cells. Endocrinology, 2008. 150(6): p. 2723-2731.

9. Li, Z., et al., SIRT6 suppresses NFATc4 expression and activation in cardiomyocyte hypertrophy. Frontiers in pharmacology, 2019. 9: p. 1519.

10. Songstad, N.T., et al., Effects of High Intensity Interval Training on Pregnant Rats, and the Placenta, Heart and Liver of Their Fetuses. PloS one, 2015. 10(11): p. e0143095.

11. Brito, V.B., et al., Exercise during pregnancy decreases doxorubicin-induced cardiotoxic effects on neonatal hearts. Toxicology, 2016. 368: p. 46-57.

12. Wisløff, U., Ø. Ellingsen, and O.J. Kemi, High-intensity interval training to maximize cardiac benefits of exercise training? Exercise and sport sciences reviews, 2009. 37(3): p. 139-146.

13. Hafstad, A., et al., High intensity interval training alters substrate utilization and reduces oxygen consumption in the heart. heart, 2011. 111: p. 1235-1241.

14. Matsuo, T., et al., Low-volume, high-intensity, aerobic interval exercise for sedentary adults: $$\\dot {V} $$ O2max, cardiac mass, and heart rate recovery. European journal of applied physiology, 2014. 114(9): p. 1963-1972.

15. Hassanieh, S. and R. Mostoslavsky, Multitasking Roles of the Mammalian Deacetylase SIRT6, in Introductory Review on Sirtuins in Biology, Aging, and Disease. 2018, Elsevier. p. 117-130.

16. Cai, Y., et al., Nmnat2 protects cardiomyocytes from hypertrophy via activation of SIRT6. FEBS letters, 2012. 586(6): p. 866-874.

17. Maksin-Matveev, A., et al., Sirtuin 6 protects the heart from hypoxic damage. Experimental cell research, 2015. 330(1): p. 81-90.

18. Huang, C.-C., et al., Effect of exercise training on skeletal muscle SIRT1 and PGC-1α expression levels in rats of different age. International journal of medical sciences, 2016. 13(4): p. 260.

19. Koltai, E., et al., Exercise alters SIRT1, SIRT6, NAD and NAMPT levels in skeletal muscle of aged rats. Mechanisms of ageing and development, 2010. 131(1): p. 21-28.

20. Høydal, M.A., et al., Running speed and maximal oxygen uptake in rats and mice: practical implications for exercise training. European Journal of Cardiovascular Prevention & Rehabilitation, 2007. 14(6): p. 753-760.

21. Rodrigues, B., et al., Maximal exercise test is a useful method for physical capacity and oxygen consumption determination in streptozotocin-diabetic rats. Cardiovascular diabetology, 2007. 6(1): p. 38.

22. Stanford, K.I., et al., Exercise before and during pregnancy prevents the deleterious effects of maternal high-fat feeding on metabolic health of male offspring. Diabetes, 2015. 64(2): p. 427-433.

23. Sheldon, R.D., et al., Gestational exercise protects adult male offspring from high-fat diet-induced hepatic steatosis. Journal of hepatology, 2016. 64(1): p. 171-178.

---

## [Decision Letter · Decision Letter 1]

25 Jun 2020

PONE-D-19-33225R1

Influence of the maternal high-intensity-interval-training on the cardiac Sirt6 and lipid profile of the adult male offspring in rats

PLOS ONE

Dear Dr. khaledi,

Thank you for submitting your manuscript to PLOS ONE. After careful consideration, we feel that it has merit but does not fully meet PLOS ONE’s publication criteria as it currently stands. Therefore, we invite you to submit a revised version of the manuscript that addresses the points raised during the review process.

Experimentally this is fine, but the conclusions do not necessarily follow from the results. The result show an impact on maternal HIIT on some parameters in the offspring but are not enough on their own to talk about general cardiovascular help. The options would either by to add more data to support such broader conclusions or to change the focus of the conclusion so as they better fit the data presented. 

We look forward to receiving your revised manuscript.

Kind regards,

Christopher Torrens

Academic Editor

PLOS ONE

Additional Editor Comments (if provided):

The paper has been improved but there are still some outstanding issues.

Table 4

Is the birthweight the mean per litter, which in turn is meaned for the group?

Looking at this table again, I don't undertstand the sex distribution values given. It seems to correspond to the number of offspring but that value is presented as a mean +/- SEM. For example the number of pups in the BP group is given as 5 +/- 1.71, which suggests quite a large variation in pups number, if so what does it mean that the split was 3:2 M:F?

Line 219: Subsection on offspring physical activity. The title here says "Maternal HIIT...could improve physical activity...". This is not really what you measured though and this becomes ambiguous. Your data show they could run longer and faster but this is not the same as being more active, which you acknowledge because the title of the section says "could improve".

Table 5 needs standardised the decimal points. The raw mass is given to three decimal places but the error is only to two, while the percentage is give to two with the error given to three places. The two measure have difference accuracies and don't need to be standardised as such, but the error for each measurement should be in the same format as the measurement it is taken.

Line 242 "Beneficial effects" This is not the you found result. The result is a change in sirt6 and IGF-2 mRNA expression in response to maternal HIIT protocols. Any interpretation of what this means, be it good or bad belongs in the discussion not the results.

Reviewers' comments:

Reviewer's Responses to Questions

**Comments to the Author**

1. If the authors have adequately addressed your comments raised in a previous round of review and you feel that this manuscript is now acceptable for publication, you may indicate that here to bypass the “Comments to the Author” section, enter your conflict of interest statement in the “Confidential to Editor” section, and submit your "Accept" recommendation.

Reviewer #1: All comments have been addressed

Reviewer #2: (No Response)

2. Is the manuscript technically sound, and do the data support the conclusions?

Reviewer #1: Yes

Reviewer #2: No

3. Has the statistical analysis been performed appropriately and rigorously? 

Reviewer #1: Yes

Reviewer #2: I Don't Know

4. Have the authors made all data underlying the findings in their manuscript fully available?

Reviewer #1: Yes

Reviewer #2: Yes

5. Is the manuscript presented in an intelligible fashion and written in standard English?

Reviewer #1: Yes

Reviewer #2: Yes

6. Review Comments to the Author

Reviewer #1: The authors have addressed my comments, I have 2 very minor concerns.

It should be made clear that “Maternal HIIT prior to and during pregnancy have beneficial effects on the heart of the next generation” are referring to the offspring – as written, it is hard to determine if the authors mean the offspring (F1) or the next generation (F2).

Line 181: Should be “vena cava” instead of Veno Cova

Reviewer #2: The authors did not perform most of the experiments I proposed in my revision. They conclude that maternal exercise may have effects on cardiovascular health, yet they fail to do studies of cardiac function.

7. PLOS authors have the option to publish the peer review history of their article (what does this mean?). If published, this will include your full peer review and any attached files.

Reviewer #1: No

Reviewer #2: No

---

## [Author Response · Author response to Decision Letter 1]

3 Jul 2020

Dear Christopher Torrens

Academic Editor

PLOS ONE

The authors would like to Thank you for giving me the opportunity to submit a revised draft of our manuscript titled [Influence of the maternal high-intensity-interval-training on the cardiac Sirt6 and lipid profile of the adult male offspring in rats, No. PONE-D-19-33225] to [PLOS ONE]. We appreciate the time and effort that you and the reviewers have dedicated to providing your feedback on our manuscript. We have been able to incorporate changes to reflect most of the suggestions provided by the reviewers. We have highlighted the changes within the manuscript. We hope that this new version of the manuscript is satisfactory for the presentation and we are thankful once again to the reviewers for their useful comments.

Sincerely yours

Neda Khaledi (Ph. D)

Corresponding Author

 

Academic editor' comments:

We truly appreciate the Editor’s suggestion, and we recognize the benefits improving the quality of the manuscript. 

 Point 1: Is the birthweight the mean per litter, which in turn is meaned for the group? 

Looking at this table again, I don't undertstand the sex distribution values given. It seems to correspond to the number of offspring but that value is presented as a mean +/- SEM. For example the number of pups in the BP group is given as 5 +/- 1.71, which suggests quite a large variation in pups number, if so what does it mean that the split was 3:2 M:F?

Response: Yes, this birthweight was reported based on the mean of all litter per group for example in the BP group is given as 6.020 ± 0.25, which this value is obtained from the meaning of pups’s birth weight of 6 mothers.

We agree with your suggestion about sex distribution and we have added SEM for it.

Point 2: Line 219: Subsection on offspring physical activity. The title here says "Maternal HIIT...could improve physical activity...". This is not really what you measured though and this becomes ambiguous. Your data show they could run longer and faster but this is not the same as being more active, which you acknowledge because the title of the section says "could improve". Response: In accordance with your suggestion, we have revised this section.

Point 3: Table 5 needs standardised the decimal points. The raw mass is given to three decimal places but the error is only to two, while the percentage is give to two with the error given to three places. The two measure have difference accuracies and don't need to be standardised as such, but the error for each measurement should be in the same format as the measurement it is taken.

Response: You have raised an important point. It was corrected.

Point 4: Line 242 "Beneficial effects" This is not the you found result. The result is a change in sirt6 and IGF-2 mRNA expression in response to maternal HIIT protocols. Any interpretation of what this means, be it good or bad belongs in the discussion not the results.

Response: Thank you for a valuable point. It was corrected.

Reviewers' comments:

Reviewer #1: Manuscript Number: PONE-D-19-33225

We are grateful to the reviewer 1 and 2 for their insightful comments on this article. 

Point 1: It should be made clear that “Maternal HIIT prior to and during pregnancy have beneficial effects on the heart of the next generation” are referring to the offspring – as written, it is hard to determine if the authors mean the offspring (F1) or the next generation (F2).

Response: Thank you for pointing this out. It was changed to offspring. 

Point 2: Line 181: Should be “vena cava” instead of Veno Cova

Response: This mistake was corrected. Thank you.

Reviewers' comments:

Reviewer #2: Manuscript Number: PONE-D-19-33225

Point 1: The authors did not perform most of the experiments I proposed in my revision. They conclude that maternal exercise may have effects on cardiovascular health, yet they fail to do studies of cardiac function.

Response: Although faced with limitation just as you said, we believe that additional experiments including direct measurements of cardiac function are very useful and will strengthen our work, but unfortunately due to the lack of funding time we are unable to investigate further at this time. accurate assessment of cardiac function, which is one of the limitations of our research, need to be known in further work and this limitation was added in the manuscript (line 299). Indeed, in accordance with your suggestion we have removed the term of cardiovascular health and have revised the manuscript.

---

## [Editor Report · Decision Letter 2]

22 Jul 2020

Influence of the maternal high-intensity-interval-training on the cardiac Sirt6 and lipid profile of the adult male offspring in rats

PONE-D-19-33225R2

Dear Dr. khaledi,

We’re pleased to inform you that your manuscript has been judged scientifically suitable for publication and will be formally accepted for publication once it meets all outstanding technical requirements.

Kind regards,

Christopher Torrens

Academic Editor

PLOS ONE
---

## [Editor Report · Acceptance letter]

24 Jul 2020

PONE-D-19-33225R2 

Influence of the Maternal High-Intensity-Interval-Training on the Cardiac *Sirt6* and Lipid Profile of the Adult Male Offspring in Rats 

Dear Dr. Khaledi:

I'm pleased to inform you that your manuscript has been deemed suitable for publication in PLOS ONE. Congratulations! Your manuscript is now with our production department. 

Kind regards, 

on behalf of

Dr. Christopher Torrens 

Academic Editor

PLOS ONE